# The Jack-in-the-Box: Pericardial Decompression Syndrome Managed by a Multidisciplinary Approach with Early Initiation of Veno-Arterial Extracorporeal Membrane Oxygenation: A Case Report

**DOI:** 10.3390/medicina60111747

**Published:** 2024-10-24

**Authors:** Carmen Orban, Tudor Borjog, Claudia Talpau, Mihaela Agapie, Angelica Bratu, Mugurel Jafal, Mihai Popescu

**Affiliations:** 1Department of Anaesthesia and Intensive Care, “Carol Davila” University of Medicine and Pharmacy, 37 Dionisie Lupu Street, 020021 Bucharest, Romania; carmen.orban@umfcd.ro (C.O.); agapiemili@yahoo.com (M.A.); jafalmugurel@yahoo.com (M.J.); mihaipopescu@umfcd.ro (M.P.); 2Department of Anaesthesia and Intensive Care, Bucharest University Emergency Hospital, 169 Independentei Street, 050098 Bucharest, Romania; claudiatalpau@yahoo.com (C.T.); angibratu@yahoo.com (A.B.)

**Keywords:** post decompression syndrome, veno-arterial extracorporeal membrane oxygenation, cardiogenic shock

## Abstract

Post decompression syndrome (PDS) is a rare and life-threatening complication of pericardiocentesis, especially after rapid drainage of large amounts of pericardial fluid. We present the case of a 21-year-old man who presented with cardiac tamponade of unknown etiology. After preoperative optimization, surgical drainage of the pericardial effusion was performed and approximately 2500 mL of fluid was released over 30 min. The patient rapidly developed hemodynamic collapse with severe biventricular dysfunction, with a left ventricle ejection fraction of 15%. Vasopressor and inotropic support were initiated with Noradrenaline and Dobutamine, further escalated to Adrenaline and Levosimendan with no improvement in clinical and hemodynamic parameters. Considering the high doses of vasoactive drugs, rescue veno-arterial extracorporeal membrane oxygenation (V-A ECMO) was started within the first 24 h. After 10 days on V-A ECMO, the cardiac function slowly recovered, and the extracorporeal mechanical support was successfully weaned. The diagnosis of paraneoplastic PDS secondary to angiosarcoma was made and the patient was successfully discharged to the ward on the 24th day. In conclusion, far from being the last option in the management of PDS, V-A ECMO deserves early consideration for securing adequate myocardial and systemic perfusion, while the cardiac function recovers, but a risk-to-benefit assessment should be made by an experienced multidisciplinary team.

## 1. Introduction

Post decompression syndrome (PDS) represents a rare, underdiagnosed, and potentially fatal complication of pericardiocentesis which was first described by Vandyke in 1983 [1]. It is defined as a paradoxical worsening of cardiovascular function and development of pulmonary edema after uncomplicated pericardial drainage of large effusions and has an estimated incidence of <5% [2]. The exact pathophysiology of PDS is unknown, but several mechanisms have been proposed [3]. Rapid removal of pericardial fluid leads to increased venous return, causing a left-sided shifting of the interventricular septum and decreased left ventricular end-systolic volume and cardiac output [4]. Also, the systemic vascular resistance increases as an adaptive response to counteract systemic hypotension. After rapid decompression, the pulmonary venous return abruptly increases, while the afterload is still high, which results in a preload-to-afterload mismatch and acute heart failure [5]. Secondly, the increased pericardial pressure is accompanied by a decline in coronary blood flow [6] leading to myocardial ischemia that can cause left ventricular myocardial stunning persisting after pericardial decompression and resulting in diastolic dysfunction [7]. Thirdly, the acute removal of sympathetic stimulus after pericardial drainage may unmask pre-existing myocardial dysfunction which was not previously revealed due to excessive circulating catecholamines with a positive chronotropic and inotropic effect [8]. The etiology of PDS is incompletely known and several risk and predictive factors have been identified including malignancy as the most common cause in up to 38% of cases [9] and female sex [2]. Other risk factors are prior radiotherapy, pre-existing cardiomyopathy with decreased systolic function, connective tissue disease and pericardial calcification [10].

Current management of PDS involves a multidisciplinary team with advanced hemodynamic monitoring and support as the hallmark of treatment. As cardiac dysfunction is mostly reversible after PDS, inotropic support and aggressive heart failure management with diuretics and vasopressors represent the main treatment options [11]. However, in refractory cases, mechanical circulatory support, such as an intra-aortic balloon pump or veno-arterial extracorporeal membrane oxygenation (V-A ECMO), may be required [9]. We present a rare case of PDS of initially unknown etiology refractory to standard medical treatment that was successfully managed with early initiation of V-A ECMO.

## 2. Case Presentation

We report the case of a 21-year-old male, without any known medical history, who was brought to the Emergency Department (ED) of a major university hospital for dyspnea, orthopnea, and cough that started two weeks prior and became progressively worse. Chest radiography showed global cardiomegaly and left pleural effusion. Subsequently, an echocardiography was performed and showed massive pericardiac effusion with swinging of the heart and cardiac tamponade with diastolic collapse of the right ventricle (Appendix A). The patient was admitted to the Intensive Care Unit (ICU) for advanced monitoring and respiratory support. On admission, the patient was alert, without any neurological deficits, spontaneously breathing with supplemental oxygen through a face mask at 8 L/min and a peripheric oxygen saturation of 98%, but he could not tolerate the supine position. Systemic blood pressure was 136/102 mmHg, and the heart rate was 136 beats/min. Physical examination showed dyspnea, polypnea, bilateral pulmonary crackles and inaudible heart sounds. He also complained of pain in the right upper abdominal quadrant. Paraclinical results revealed a prothrombin time of 20.1 s, mild hepatocytolysis (alanine transaminase of 284 U/L, aspartate aminotransferase of 347 U/L), prehepatic jaundice (total bilirubin of 2.67 mg/dL with conjugated bilirubin of 0.82 mg/dL) and moderate hyponatremia of 127 mmol/L. Severe acute respiratory syndrome coronavirus 2 (SARS-CoV-2) test was negative. The electrocardiography showed sinus tachycardia, ventricular bigeminy, right bundle branch block and low-voltage QRS complexes.

Firstly, left pleural drainage of approximately 1000 mL of serous fluid was performed, after which the patient was transferred to the operating room. After a rapid sequence induction with Etomidate 200 mcg/kg and Rocuronium 1 mg/kg, pericardial drainage through a subxiphoidian approach was performed and a large amount of approximately 2500 mL of sero-hematic pericardial fluid was incrementally evacuated over thirty minutes. Shortly after that, the patient became severely hemodynamically unstable and vasopressor support with Noradrenaline was initiated (up to 0.75 mcg/kg/min). A second echocardiography revealed severe biventricular dysfunction, with a left ventricle ejection fraction (LVEF) of 10–15% (Appendix A). Due to persistent hemodynamic collapse, inotropic support with Dobutamine 10 mcg/kg/min was started with further addition of Adrenaline 0.06 mcg/kg/min and Levosimendan 0.1 mcg/kg/min. Invasive hemodynamic monitoring by transpulmonary thermodilution demonstrated a decreased cardiac index of 1.0 L/min/m^2^, global end-diastolic index of 729 mL/m^2^, extravascular lung water index of 15 mL/kg and an indexed systemic vascular resistance of 5400 Dyn·s/cm^5^·m^2^.

Based on the clinical and paraclinical findings, as well as on the large amount of pericardial fluid that had been drained too fast, suspicion of PDS was raised. A contrast-enhanced computer tomography of the chest and abdomen was performed and showed an anterior mediastinal mass, with areas of necrosis, invading both atria and the interatrial septum and surrounding the right pulmonary artery, the superior vena cava, the aortic arch and the trachea (Appendix A). A multidisciplinary team consisting of a cardiologist, oncologist, anesthesiologists, intensivists and thoracic and cardiovascular surgeons further evaluated the best treatment option. Considering the reversibility of the PDS and the poor response to inotropic support up to that point, V-A ECMO was rapidly initiated. The right femoral vein and left femoral artery were cannulated and V-A ECMO was started with an initial flow of 2.2 L/min, 5300 rotations/min and a sweep gas flow of 4 L/min.

Due to the development of acute kidney injury and fluid overload, continuous renal replacement therapy (CRRT) was also initiated on the 2nd ICU day. Of note, urine output during the 1st ICU day was 2450 mL and decreased to oliguria during the following day. The clinical course was complicated on the 7th ICU day by infection of the necrotic mediastinal mass and septic shock. Broad-spectrum antibiotics were initiated with Meropenem (1 g intravenously at 8 h intervals), Tigecycline (100 mg intravenous loading dose, then 50 mg intravenously at 12 h intervals) and Caspofungin (70 mg intravenous loading dose infused in 1 h, then 50 mg intravenously per day). To manage the severe systemic inflammatory syndrome, hemoadsorption with Cytosorb was initiated for 6 h. The clinical course was favorable with remission of septic shock, but CRRT was continued for the management of acute kidney injury. Dynamics of paraclinical results, hemodynamic parameters and vasopressor support are presented in Table 1.

Under V-A ECMO, the cardiac function and left ventricle ejection fraction slowly improved and the inotropic support and V-A ECMO parameters were progressively weaned off after 10 days. The most important echocardiographic findings in relation to V-A ECMO initiation and weaning are summarized in Table 2.

On the 10th ICU day, sedation was discontinued, and the patient became fully alert without neurological deficits, so he could be weaned from invasive mechanical ventilation to high-flow oxygen therapy. He remained hemodynamically stable under CRRT until the recovery of the renal function on the 16th ICU day. A medical rehabilitation program was started, but with minimal recovery of the functional status because of the recurrence of the pericardial effusion. After 18 days in the ICU, surgical reintervention to biopsy the mediastinal mass was performed. At the same time, a pleuro-pericardial window was carried out. The diagnosis of angiosarcoma was confirmed by histopathologic and immunohistochemistry examination. Postoperatively, the patient remained hemodynamically stable and could be weaned off early from respiratory support and was discharged from the ICU after 24 days to be fully assessed by the oncology team in the view of starting chemotherapy.

A timeline containing the main therapeutic interventions and patient course is presented in Figure 1.

## 3. Discussion

PDS is a relatively rare life-threatening complication of pericardiocentesis with high mortality rates unless appropriate advanced cardiovascular support is applied. The exact epidemiology of PDS is unknown but most patient cohorts report an incidence between 4.8% and 11% [12,13]. However, in a large consecutive series of 1067 patients who underwent pericardiocentesis, only one episode of PDS was described [14]. An important risk factor for the development of PDS is the rapid drainage of large quantities of pericardial fluid, either surgically or percutaneously. In a comprehensive analysis of 35 patients with PDS by Pradhan et al. [15], the minimum and maximum amount of fluid drained was 450 and 2100 mL, respectively. To date, there are no clear recommendations regarding the amount and rate of removal of pericardial fluid to prevent PDS, but from a pathophysiological point of view, we should remove the amount of fluid that results in the resolution of cardiac tamponade without exceeding one liter. Further, fluid can subsequently be drained by placing an indwelling pericardial catheter for a slow, gradual decompression [16]. The indwelling catheter can be removed when the pericardial drainage is under 20–30 mL/day [17].

Our case presents a rare situation of cardiac tamponade in an apparently healthy young patient. We faced a significant diagnostic and therapeutic challenge due to the urgency of the clinical presentation of acute respiratory failure in a patient without identifiable risk factors. Unfortunately, the patient was admitted directly to the ICU and urgent thoracocentesis and pericardiocentesis were more important as first-line therapy than performing a comprehensive computer tomography (CT) to rule out malignancy as an underlying cause. Also, as the patient could not tolerate lying flat for the CT and, due to the risk of cardiovascular collapse, he was taken directly to the operating room. However, due to the national epidemiological situation at that time, both rapid and polymerized chain reaction testing ruled out SARS-CoV-2 infection as one of the main causes of cardiac tamponade in young adults [18,19]. Cardiac Magnetic Resonance (CMR) has been recently recommended as a diagnostic tool for pericardial disease as it can non-invasively characterize both the myocardium and mediastinal masses [20]. However, we could not perform CMR imaging in the beginning due to the urgency of the case. A biopsy was considered to be the best diagnostic approach at the time and was performed shortly after V-A ECMO was successfully weaned off, and due to the sternal wires, we could not perform the CMR imaging afterwards.

Most cases of PDS present with either uni- or biventricular dysfunction with or without pulmonary edema [21,22]. However, some case reports presented only with pulmonary edema without any left ventricular failure [23]. Another cause of paradoxical cardiac decompensation in patients with mediastinal masses and cardiac tamponade after pericardial drainage is the compression of the right chambers by the tumor itself with further worsening of the obstructive shock. In such cases, the pericardial effusion has a beneficial effect, decreasing the compressive mass effect of the tumor on the cardiac chambers, and reinfusion of normal saline in the pericardium could be attempted [24]. However, in our case, transthoracic echocardiography showed no compression of the right atrium and ventricle after evacuation of the pericardial effusion. Our case presented with severe left ventricular dysfunction and pulmonary edema as demonstrated by both echocardiographic and invasive hemodynamic monitoring parameters for which we started inotropic support but without a significant clinical and hemodynamic improvement. Moreover, we noted a further increase in systemic vascular resistance, thus maintaining a vicious circle of cardiogenic shock. Because of the failure of inotropic support, V-A ECMO was initiated via the femoral approach. This option for the early initiation of extracorporeal support has been proposed as a rescue therapy by different case reports. Veiras et al. [25] reported a case of a 68-year-old woman with cardiovascular arrest after pericardiocentesis in whom V-A ECMO was applied on the operating table with an initial improvement in cardiac function and a favorable outcome. In another case of PDS in a 16-year-old child with Hodgkin lymphoma, the patient required early ECMO for cardiovascular support that could be weaned off after 9 days [26]. From these case reports, as well as others [27,28], we considered that V-A ECMO may represent a life-saving therapy in severe cases unresponsive to standard medical therapy. However, this strategy is not without risks and serious complications have been reported [29]. Thus, one must balance the risks of the procedures with the severity of cardiac dysfunction to establish the best approach for a better outcome.

## 4. Conclusions

Pericardiocentesis can sometimes prove fatal in cases complicated by PDS. The only proven method by which its occurrence can be limited is by minimizing the amount of pericardial fluid drained at a time, so percutaneous drainage should be favored over surgical drainage. Rather than being an all-or-none phenomenon, PDS exhibits variable severity but is most often reversible. Hence, a timely and exhaustive therapeutic response must be endorsed to ensure the best outcomes. Far from being the last option in the management of PDS, V-A ECMO deserves early consideration for securing adequate myocardial and systemic perfusion while the cardiac function recovers, as in the case of our patient.

## Figures and Tables

**Figure 1 medicina-60-01747-f001:**
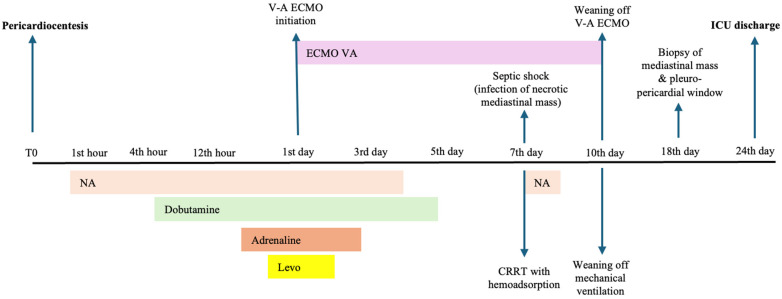
Timeline representation of the main therapeutic interventions during Intensive Care Unit stay (T0 denotes the time of pericardiocentesis). Legend: NA—Noradrenaline infusion; Dobutamine—Dobutamine infusion; Adrenaline—Adrenaline infusion; Levo—Levosimendan infusion; CRRT—continuous renal replacement therapy; V-A ECMO—veno-arterial extracorporeal membrane oxygenation; ICU—Intensive Care Unit.

**Table 1 medicina-60-01747-t001:** Dynamics of clinical, paraclinical and treatment parameters throughout ICU stay.

Parameter	ICU Admission	ICU Day 1	ICU Day 3	ICU Day 5	ICU Day 7	ICU Day 10	ICU Discharge
Cardiovascular parameters							
Noradrenaline (mcg/kg/min)	0.1	0.75	0.5	-	-	-	-
Adrenaline (mcg/kg/min)	-	0.06	-	-	-	-	-
Dobutamine (mcg/kg/min)	-	10	6	-	-	-	-
Levosimendan (mcg/kg/min)	-	0.1	-	-	-	-	-
Urine output (mL/kg/h)	1.35	1.36	<0.1	0.1	<0.1	<0.1	1.66
Mean HR (beats/min)	136	108	106	108	104	124	109
BP: Sys/Dia/Mean (mmHg)	136/102/113	60/40/46	98/52/67	90/52/64	112/73/86	106/50/68	128/73/91
EF-TTE (%)	NA	15	15	25	30	35	45
CI (L/min/m^2^)	NA	1.0	1.8	NA	NA	NA	NA
ELWI (mL/kg)	NA	15	16	NA	NA	NA	NA
GEDI (mL/m^2^)	NA	729	620	NA	NA	NA	NA
SVRI (dyn·s·cm^−5^·m^2^)	NA	5400	2800	NA	NA	NA	NA
Paraclinical tests							
ALT (U/L)	284	400	606	374	140	33	14
AST (U/L)	347	569	607	170	50	30	40
TB (mg/dL)	2.67	2.16	2.84	3.45	2.54	2.9	1.84
WBCs (×10^3^/µL)	10.6	11	13.1	10.6	9.5	8.9	7.4
Hb (mg/dL)	13.3	12.3	11.6	10.7	11.9	8.9	8.9
PLTs (×10^3^/µL)	329	265	224	169	176	102	587
SCr (mg/dL)	1.04	1.5	1.45	1.41	1.02	3.83	1.17
pH	7.40	7.26	7.46	7.44	7.38	7.35	7.45
PaO_2_/FiO_2_	304	194	322	356	280	255	480
Lactate (mmol/L)	2.80	2.29	1.58	1.35	0.72	1.02	1.32
HCO_3_ (mmol/L)	22.0	17.8	21.0	22.2	23.4	19.0	22.7

Legend: ICU—Intensive Care Unit; HR—heart rate; BP—blood pressure; Sys—systolic; Dia—diastolic; EF—ejection fraction; TTE—transthoracic echocardiography; CI—cardiac index; ELWI—extravascular lung water index; GEDI—global end-diastolic volume index; SVRI—systemic vascular resistance index; ALT—alanine transaminase; AST—aspartate aminotransferase; TB—total bilirubin; WBCs—white blood cells; Hb—hemoglobin; PLTs—platelets; SCr—serum creatinine; PaO_2_—arterial oxygen partial pressure; FiO_2_—inspired oxygen fraction. NA—not available.

**Table 2 medicina-60-01747-t002:** Echocardiographic parameters in relation to V-A ECMO.

Parameter	After Pericardial Drainage	At V-A ECMO Initiation	5th Day on V-A ECMO	7th Day on V-A ECMO	After V-A ECMO Weaning
LVEF (%)	10	10	15	30	40
TAPSE (mm)	6	6	9	12	16
LVOT-VTI (m/s)	9	8	NA	11.5	16
IVC (mm)	30	44	22	22	18
Mitral regurgitation grade	II	II	II	I	I

Legend: LVEF—left ventricle ejection fraction; TAPSE—tricuspid annular plane systolic excursion; LVOT-VTI—left ventricle outflow tract velocity time integral; IVC—inferior vena cava; V-A ECMO—veno-arterial extracorporeal membrane oxygenation. NA—not available.

## Data Availability

Appendix A can be provided by the corresponding author.

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
