# Peer review of "The Jack-in-the-Box: Pericardial Decompression Syndrome Managed by a Multidisciplinary Approach with Early Initiation of Veno-Arterial Extracorporeal Membrane Oxygenation: A Case Report"

_medicina, 2024, doi:10.3390/medicina60111747_

Round 1

Reviewer 1 Report

Comments and Suggestions for Authors

Authors of this case report investigated a patient with pericardial decompression syndrome (PDS) and valuated the outcome after intensive care unit care. Veno-arterial extracorporeal membrane oxygenation (V-E ECMO) was adopted in order to facilitate myocardial function recovery.   The paper underlined the physiopathology of the syndrome and described all the diagnostic and therapeutic measures performed. It also summarized literature produced in last years. Authors highlighted the importance of pericardiocentesis in correct heart chamber decompression. Other papers were published quite recently regarding the same topic. Anyway it remains a major concern in modern heart surgery clinics.

I would suggest to take into consideration the following comments in order to improve the overall quality of the manuscript:

1)      In abstract form should be reported in summary the content of lines 197-198 of discussion. It is important because not always is possible to evacuate a pericardial effusion with pericardiocentesis.

2)      In case presentation section it should be better to introduce values of urine output in hospital admission (value reported in table 1).

3)      Contents from line 150 to 161 and from line 173 to 180 should be put into introduction section.

4)      In case presentation section it should be underlined that a large amount of liquid was drained too quickly. This element is correctly described in discussion section.

5)      It should be introduced in the paper other ultrasound scan parameters (before treatment, during V-E ECMO, and after recovery). These should be summarized in a new table.

6)      English should be improved.

Comments on the Quality of English Language

I suggest an English revision  by mother tongue speakers

Author Response

Dear Reviewer,

Thank you very much for critically reading our manuscript and the suggestions provided that significantly improved the quality of our paper. We will further address the issues you have raised in a systematic matter.

Q1. In abstract form should be reported in summary the content of lines 197-198 of discussion. It is important because not always is possible to evacuate a pericardial effusion with pericardiocentesis.

R1. We have included the summary of the aforementioned lines in the abstract for a better understanding of the issue of such cases. Also, we have shortened the abstract as recommended by the 2ndreviewer.  

Q2. In the case presentation section it should be better to introduce values of urine output in hospital admission (value reported in table 1).

R2. We have introduced the values for urine output at hospital admission in the case presentation as recommended by the reviewer.

Q3. Contents from line 150 to 161 and from line 173 to 180 should be put into introduction section.

R3. The reviewer is right, contents from lines 150 to 161 were best fitted in the introduction section after the pathophysiological considerations of PDS, however, based on the other reviewer’s recommendations and the natural flow of reading the article we have considered that is better to keep lines 173 to 180 in the discussion section as it comes (to discuss) the clinical presentation of PDS in comparison to our own.   

Q4. In the case presentation section, it should be underlined that a large amount of liquid was drained too quickly. This element is correctly described in the discussion section.

R4. We have underlined this aspect in two sections of the case presentation so that it is clear to the reader that a large amount of pericardial fluid was drained too fast, and this was the main risk factor for PDS. Also, as the reviewer pointed out, the topic of speed of drainage vs. PDS has been described in the discussion section.   

Q5. It should be introduced in the paper other ultrasound scan parameters (before treatment, during V-E ECMO, and after recovery). These should be summarized in a new table.

R5. We have introduced a new table containing TTE parameters in the case presentation (table 2) as recommended by the reviewer.

Reviewer 2 Report

Comments and Suggestions for Authors

We thank the authors for submitting their manuscript to our journal. The article focuses on Pericardial Decompression Syndrome managed by a multidisciplinary approach with early initiation of veno-arterial extracorporeal membrane oxygenation: a case report. The following minor revisions are required:

1) The abstract length should be reduced to less than 250 words to enhance conciseness and clarity.

2) It is advisable to create a timetable and a timeline of the diagnostic and therapeutic interventions performed to make the patient's care pathway more comprehensible for readers.

3) We suggest including and discussing echocardiographic images referenced in the text, in addition to the video in the supplementary materials, to provide a visual context for the findings presented.

4) Similarly, incorporating and commenting on CT images mentioned in the text, alongside the supplementary video, will enhance the manuscript's depth and utility.

5) Please indicate whether a follow-up assessment using Cardiac Magnetic Resonance (CMR) was recommended. CMR offers optimal imaging of the pericardium and facilitates non-invasive tissue characterization of both the myocardium and mediastinal masses. It is capable of diagnosing both acquired and congenital pericardial pathologies, as outlined by Trimarchi et al. (doi:10.1093/ehjcr/ytae200).

These revisions will improve the quality and accessibility of the manuscript for our readership.

Author Response

Dear Reviewer,

Thank you very much for critically reading our manuscript and the suggestions provided that significantly improved the quality of our paper. We will further address the issues you have raised in a systematic matter.

Q1. The abstract length should be reduced to less than 250 words to enhance conciseness and clarity.

R1. We have shortened the abstract to the minimal necessary information and it has now been reduced to 207 words as indicated.

Q2. It is advisable to create a timetable and a timeline of the diagnostic and therapeutic interventions performed to make the patient's care pathway more comprehensible for readers.

R2. We have created a timeline that summarizes the most important aspects of patient care so that the reader can easily follow them. This has now been included in the article as Figure 1 as suggested.    

Q3. We suggest including and discussing echocardiographic images referenced in the text, in addition to the video in the supplementary materials, to provide a visual context for the findings presented.

R3. We have included a more detailed interpretation of the supplementary materials. This can be found at the end of the article highlighted in yellow.

Q4. Similarly, incorporating and commenting on CT images mentioned in the text, alongside the supplementary video, will enhance the manuscript's depth and utility.

R4. We have included a more detailed interpretation of the supplementary materials. This can be found at the end of the article highlighted in yellow.

Q5. Please indicate whether a follow-up assessment using Cardiac Magnetic Resonance (CMR) was recommended. CMR offers optimal imaging of the pericardium and facilitates non-invasive tissue characterization of both the myocardium and mediastinal masses. It is capable of diagnosing both acquired and congenital pericardial pathologies, as outlined by Trimarchi et al. (doi:10.1093/ehjcr/ytae200).

R5. Indeed, CMR has been recently recommended as a diagnostic tool for pericardial disease as it can non-invasively characterize both the myocardium and mediastinal masses. However, we could not perform CMR imaging in the beginning due to the urgency of the case. A biopsy was considered to be the best diagnostic approach at the time and was performed shortly after V-A ECMO was successfully weaned off and, due to the sternal wires, we could not perform the CMR imaging afterwards. This has been introduced in the discussion section as recommended by the reviewer.